# First-Order Adaptive Sample Size Methods to Reduce Complexity of Empirical Risk Minimization

**Aryan Mokhtari**
University of Pennsylvania
aryanm@seas.upenn.edu

**Alejandro Ribeiro**
University of Pennsylvania
aribeiro@seas.upenn.edu

## Abstract

This paper studies empirical risk minimization (ERM) problems for large-scale datasets and incorporates the idea of adaptive sample size methods to improve the guaranteed convergence bounds for first-order stochastic and deterministic methods. In contrast to traditional methods that attempt to solve the ERM problem corresponding to the full dataset directly, adaptive sample size schemes start with a small number of samples and solve the corresponding ERM problem to its statistical accuracy. The sample size is then grown geometrically – e.g., scaling by a factor of two – and use the solution of the previous ERM as a warm start for the new ERM. Theoretical analyses show that the use of adaptive sample size methods reduces the overall computational cost of achieving the statistical accuracy of the whole dataset for a broad range of deterministic and stochastic first-order methods. The gains are specific to the choice of method. When particularized to, e.g., accelerated gradient descent and stochastic variance reduce gradient, the computational cost advantage is a logarithm of the number of training samples. Numerical experiments on various datasets confirm theoretical claims and showcase the gains of using the proposed adaptive sample size scheme.

## 1 Introduction

Finite sum minimization (FSM) problems involve objectives that are expressed as the sum of a typically large number of component functions. Since evaluating descent directions is costly, it is customary to utilize stochastic descent methods that access only one of the functions at each iteration. When considering first order methods, a fitting measure of complexity is the total number of gradient evaluations that are needed to achieve optimality of order $\epsilon$. The paradigmatic deterministic gradient descent (GD) method serves as a naive complexity upper bound and has long been known to obtain an $\epsilon$-suboptimal solution with $\mathcal{O}(N\kappa \log(1/\epsilon))$ gradient evaluations for an FSM problem with $N$ component functions and condition number $\kappa$ [13]. Accelerated gradient descent (AGD) [14] improves the computational complexity of GD to $\mathcal{O}(N\sqrt{\kappa} \log(1/\epsilon))$, which is known to be the optimal bound for deterministic first-order methods [13]. In terms of stochastic optimization, it has been only recently that linearly convergent methods have been proposed. Stochastic averaging gradient [15, 8], stochastic variance reduction [10], and stochastic dual coordinate ascent [17, 18], have all been shown to converge to $\epsilon$-accuracy at a cost of $\mathcal{O}((N+\kappa) \log(1/\epsilon))$ gradient evaluations. The accelerating catalyst framework in [11] further reduces complexity to $\mathcal{O}((N+\sqrt{N\kappa}) \log(\kappa) \log(1/\epsilon))$ and the works in [1] and [7] to $\mathcal{O}((N + \sqrt{N\kappa}) \log(1/\epsilon))$. The latter matches the upper bound on the complexity of stochastic methods [20].

Perhaps the main motivation for studying FSM is the solution of empirical risk minimization (ERM) problems associated with a large training set. ERM problems are particular cases of FSM, but they do have two specific qualities that come from the fact that ERM is a proxy for statistical loss minimization. The first property is that since the empirical risk and the statistical loss have different

minimizers, there is no reason to solve ERM beyond the expected difference between the two objectives. This so-called *statistical accuracy* takes the place of $\epsilon$ in the complexity orders of the previous paragraph and is a constant of order $\mathcal{O}(1/N^\alpha)$ where $\alpha$ is a constant from the interval $[0.5, 1]$ depending on the regularity of the loss function; see Section 2. The second important property of ERM is that the component functions are drawn from a common distribution. This implies that if we consider subsets of the training set, the respective empirical risk functions are not that different from each other and, indeed, their differences are related to the statistical accuracy of the subset.

The relationship of ERM to statistical loss minimization suggests that ERM problems have more structure than FSM problems. This is not exploited by most existing methods which, albeit used for ERM, are in fact designed for FSM. The goal of this paper is to exploit the relationship between ERM and statistical loss minimization to achieve lower overall computational complexity for a broad class of first-order methods applied to ERM. The technique we propose uses subsamples of the training set containing $n \leq N$ component functions that we grow geometrically. In particular, we start by a small number of samples and minimize the corresponding empirical risk added by a regularization term of order $V_n$ up to its statistical accuracy. Note that, based on the first property of ERM, the added adaptive regularization term does not modify the required accuracy while it makes the problem strongly convex and improves the problem condition number. After solving the subproblem, we double the size of the training set and use the solution of the problem with $n$ samples as a warm start for the problem with $2n$ samples. This is a reasonable initialization since based on the second property of ERM the functions are drawn from a joint distribution, and, therefore, the optimal values of the ERM problems with $n$ and $2n$ functions are not that different from each other. The proposed approach succeeds in exploiting the two properties of ERM problems to improve complexity bounds of first-order methods. In particular, we show that to reach the statistical accuracy of the full training set the adaptive sample size scheme reduces the overall computational complexity of a broad range of first-order methods by a factor of $\log(N^\alpha)$. For instance, the overall computational complexity of adaptive sample size AGD to reach the statistical accuracy of the full training set is of order $\mathcal{O}(N\sqrt{\kappa})$ which is lower than $\mathcal{O}((N\sqrt{\kappa})\log(N^\alpha))$ complexity of AGD.

**Related work.** The adaptive sample size approach was used in [6] to improve the performance of the SAGA method [8] for solving ERM problems. In the dynamic SAGA (DynaSAGA) method in [6], the size of training set grows at each iteration by adding two new samples, and the iterates are updated by a single step of SAGA. Although DynaSAGA succeeds in improving the performance of SAGA for solving ERM problems, it does not use an adaptive regularization term to tune the problem condition number. Moreover, DynaSAGA only works for strongly convex functions, while in our proposed scheme the functions are convex (not necessarily strongly convex). The work in [12] is the most similar work to this manuscript. The Ada Newton method introduced in [12] aims to solve each subproblem within its statistical accuracy with a single update of Newton's method by ensuring that iterates always stay in the quadratic convergence region of Newton's method. Ada Newton reaches the statistical accuracy of the full training in almost two passes over the dataset; however, its computational complexity is prohibitive since it requires computing the objective function Hessian and its inverse at each iteration.

## 2   Problem Formulation

Consider a decision vector $\mathbf{w} \in \mathbb{R}^p$, a random variable $Z$ with realizations $z$ and a convex loss function $f(\mathbf{w}; z)$. We aim to find the optimal argument that minimizes the optimization problem

$$\mathbf{w}^* := \underset{\mathbf{w}}{\operatorname{argmin}} \, L(\mathbf{w}) = \underset{\mathbf{w}}{\operatorname{argmin}} \, \mathbf{E}_Z[f(\mathbf{w}, Z)] = \underset{\mathbf{w}}{\operatorname{argmin}} \int_{\mathbf{Z}} f(\mathbf{w}, Z) P(dz), \qquad (1)$$

where $L(\mathbf{w}) := \mathbf{E}_Z[f(\mathbf{w}, Z)]$ is defined as the expected loss, and $P$ is the probability distribution of the random variable $Z$. The optimization problem in (1) cannot be solved since the distribution $P$ is unknown. However, we have access to a training set $\mathcal{T} = \{z_1, \ldots, z_N\}$ containing $N$ independent samples $z_1, \ldots, z_N$ drawn from $P$, and, therefore, we attempt to minimize the empirical loss associated with the training set $\mathcal{T} = \{z_1, \ldots, z_N\}$, which is equivalent to minimizing the problem

$$\mathbf{w}_n^\dagger := \underset{\mathbf{w}}{\operatorname{argmin}} \, L_n(\mathbf{w}) = \underset{\mathbf{w}}{\operatorname{argmin}} \, \frac{1}{n} \sum_{i=1}^{n} f(\mathbf{w}, z_i), \qquad (2)$$

for $n = N$. Note that in (2) we defined $L_n(\mathbf{w}) := (1/n)\sum_{i=1}^{n} f(\mathbf{w}, z_i)$ as the empirical loss.

There is a rich literature on bounds for the difference between the expected loss $L$ and the empirical loss $L_n$ which is also referred to as *estimation error* [4, 3]. We assume here that there exists a constant $V_n$, which depends on the number of samples $n$, that upper bounds the difference between the expected and empirical losses for all $\mathbf{w} \in \mathbb{R}^p$

$$\mathbb{E}\left[\sup_{\mathbf{w}\in\mathbb{R}^p} |L(\mathbf{w}) - L_n(\mathbf{w})|\right] \leq V_n, \tag{3}$$

where the expectation is with respect to the choice of the training set. The celebrated work of Vapnik in [19, Section 3.4] provides the upper bound $V_n = \mathcal{O}(\sqrt{(1/n)\log(1/n)})$ which can be improved to $V_n = \mathcal{O}(\sqrt{1/n})$ using the chaining technique (see, e.g., [5]). Bounds of the order $V_n = O(1/n)$ have been derived more recently under stronger regularity conditions that are not uncommon in practice, [2, 9, 4]. In this paper, we report our results using the general bound $V_n = O(1/n^\alpha)$ where $\alpha$ can be any constant form the interval $[0.5, 1]$.

The observation that the optimal values of the expected loss and empirical loss are within a $V_n$ distance of each other implies that there is no gain in improving the optimization error of minimizing $L_n$ beyond the constant $V_n$. In other words, if we find an approximate solution $\mathbf{w}_n$ such that the optimization error is bounded by $L_n(\mathbf{w}_n) - L_n(\mathbf{w}_n^\dagger) \leq V_n$, then finding a more accurate solution to reduce the optimization error is not beneficial since the overall error, i.e., the sum of estimation and optimization errors, does not become smaller than $V_n$. Throughout the paper we say that $\mathbf{w}_n$ solves the ERM problem in (2) to within its statistical accuracy if it satisfies $L_n(\mathbf{w}_n) - L_n(\mathbf{w}_n^\dagger) \leq V_n$.

We can further leverage the estimation error to add a regularization term of the form $(cV_n/2)\|\mathbf{w}\|^2$ to the empirical loss to ensure that the problem is strongly convex. To do so, we define the regularized empirical risk $R_n(\mathbf{w}) := L_n(\mathbf{w}) + (cV_n/2)\|\mathbf{w}\|^2$ and the corresponding optimal argument

$$\mathbf{w}_n^* := \underset{\mathbf{w}}{\operatorname{argmin}} R_n(\mathbf{w}) = \underset{\mathbf{w}}{\operatorname{argmin}} L_n(\mathbf{w}) + \frac{cV_n}{2}\|\mathbf{w}\|^2, \tag{4}$$

and attempt to minimize $R_n$ with accuracy $V_n$. Since the regularization in (4) is of order $V_n$ and (3) holds, the difference between $R_n(\mathbf{w}_n^*)$ and $L(\mathbf{w}^*)$ is also of order $V_n$ – this is not immediate as it seems; see [16]. Thus, the variable $\mathbf{w}_n$ solves the ERM problem in (2) to within its statistical accuracy if it satisfies $R_n(\mathbf{w}_n) - R_n(\mathbf{w}_n^*) \leq V_n$. It follows that by solving the problem in (4) for $n = N$ we find $\mathbf{w}_N^*$ that solves the expected risk minimization in (1) up to the statistical accuracy $V_N$ of the full training set $\mathcal{T}$. In the following section we introduce a class of methods that solve problem (4) up to its statistical accuracy faster than traditional deterministic and stochastic descent methods.

## 3 Adaptive Sample Size Methods

The empirical risk minimization (ERM) problem in (4) can be solved using state-of-the-art methods for minimizing strongly convex functions. However, these methods never exploit the particular property of ERM that the functions are drawn from the same distribution. In this section, we propose an *adaptive sample size* scheme which exploits this property of ERM to improve the convergence guarantees for traditional optimization method to reach the statistical accuracy of the full training set. In the proposed adaptive sample size scheme, we start by a small number of samples and solve its corresponding ERM problem with a specific accuracy. Then, we double the size of the training set and use the solution of the previous ERM problem – with half samples – as a warm start for the new ERM problem. This procedure keeps going until the training set becomes identical to the given training set $\mathcal{T}$ which contains $N$ samples.

Consider the training set $\mathcal{S}_m$ with $m$ samples as a subset of the full training $\mathcal{T}$, i.e., $\mathcal{S}_m \subset \mathcal{T}$. Assume that we have solved the ERM problem corresponding to the set $\mathcal{S}_m$ such that the approximate solution $\mathbf{w}_m$ satisfies the condition $\mathbb{E}[R_m(\mathbf{w}_m) - R_m(\mathbf{w}_m^*)] \leq \delta_m$. Now the next step in the proposed adaptive sample size scheme is to double the size of the current training set $\mathcal{S}_m$ and solve the ERM problem corresponding to the set $\mathcal{S}_n$ which has $n = 2m$ samples and contains the previous set, i.e., $\mathcal{S}_m \subset \mathcal{S}_n \subset \mathcal{T}$.

We use $\mathbf{w}_m$ which is a proper approximate for the optimal solution of $R_m$ as the initial iterate for the optimization method that we use to minimize the risk $R_n$. This is a reasonable choice if the optimal arguments of $R_m$ and $R_n$ are close to each other, which is the case since samples are drawn from

---

**Algorithm 1** Adaptive Sample Size Mechanism

---

1: **Input:** Initial sample size $n = m_0$ and argument $\mathbf{w}_n = \mathbf{w}_{m_0}$ with $\|\nabla R_n(\mathbf{w}_n)\| \leq (\sqrt{2c})V_n$

2: **while** $n \leq N$ **do** {main loop}

3:     Update argument and index: $\mathbf{w}_m = \mathbf{w}_n$ and $m = n$.

4:     Increase sample size: $n = \min\{2m, N\}$.

5:     Set the initial variable: $\tilde{\mathbf{w}} = \mathbf{w}_m$.

6:     **while** $\|\nabla R_n(\tilde{\mathbf{w}})\| > (\sqrt{2c})V_n$ **do**

7:        Update the variable $\tilde{\mathbf{w}}$: Compute $\tilde{\mathbf{w}} = \text{Update}(\tilde{\mathbf{w}}, \nabla R_n(\tilde{\mathbf{w}}))$

8:     **end while**

9:     Set $\mathbf{w}_n = \tilde{\mathbf{w}}$.

10: **end while**

---

a fixed distribution $\mathcal{P}$. Starting with $\mathbf{w}_m$, we can use first-order descent methods to minimize the empirical risk $R_n$. Depending on the iterative method that we use for solving each ERM problem we might need different number of iterations to find an approximate solution $\mathbf{w}_n$ which satisfies the condition $\mathbb{E}[R_n(\mathbf{w}_n) - R_n(\mathbf{w}_n^*)] \leq \delta_n$. To design a comprehensive routine we need to come up with a proper condition for the required accuracy $\delta_n$ at each phase.

In the following proposition we derive an upper bound for the expected suboptimality of the variable $\mathbf{w}_m$ for the risk $R_n$ based on the accuracy of $\mathbf{w}_m$ for the previous risk $R_m$ associated with the training set $\mathcal{S}_m$. This upper bound allows us to choose the accuracy $\delta_m$ efficiently.

**Proposition 1.** *Consider the sets $\mathcal{S}_m$ and $\mathcal{S}_n$ as subsets of the training set $\mathcal{T}$ such that $\mathcal{S}_m \subset \mathcal{S}_n \subset \mathcal{T}$, where the number of samples in the sets $\mathcal{S}_m$ and $\mathcal{S}_n$ are $m$ and $n$, respectively. Further, define $\mathbf{w}_m$ as an $\delta_m$ optimal solution of the risk $R_m$ in expectation, i.e., $\mathbb{E}[R_m(\mathbf{w}_m) - R_m^*] \leq \delta_m$, and recall $V_n$ as the statistical accuracy of the training set $\mathcal{S}_n$. Then the empirical risk error $R_n(\mathbf{w}_m) - R_n(\mathbf{w}_n^*)$ of the variable $\mathbf{w}_m$ corresponding to the set $\mathcal{S}_n$ in expectation is bounded above by*

$$\mathbb{E}[R_n(\mathbf{w}_m) - R_n(\mathbf{w}_n^*)] \leq \delta_m + \frac{2(n-m)}{n}\left(V_{n-m} + V_m\right) + 2\left(V_m - V_n\right) + \frac{c(V_m - V_n)}{2}\|\mathbf{w}^*\|^2. \quad (5)$$

*Proof.* See Section 7.1 in the supplementary material. □

The result in Proposition 1 characterizes the sub-optimality of the variable $\mathbf{w}_m$, which is an $\delta_m$ sub-optimal solution for the risk $R_m$, with respect to the empirical risk $R_n$ associated with the set $\mathcal{S}_n$. If we assume that the statistical accuracy $V_n$ is of the order $\mathcal{O}(1/n^\alpha)$ and we double the size of the training set at each step, i.e., $n = 2m$, then the inequality in (5) can be simplified to

$$\mathbb{E}[R_n(\mathbf{w}_m) - R_n(\mathbf{w}_n^*)] \leq \delta_m + \left[2 + \left(1 - \frac{1}{2^\alpha}\right)\left(2 + \frac{c}{2}\|\mathbf{w}^*\|^2\right)\right]V_m. \quad (6)$$

The expression in (6) formalizes the reason that there is no need to solve the sub-problem $R_m$ beyond its statistical accuracy $V_m$. In other words, even if $\delta_m$ is zero the expected sub-optimality will be of the order $\mathcal{O}(V_m)$, i.e., $\mathbb{E}[R_n(\mathbf{w}_m) - R_n(\mathbf{w}_n^*)] = \mathcal{O}(V_m)$. Based on this observation, The required precision $\delta_m$ for solving the sub-problem $R_m$ should be of the order $\delta_m = \mathcal{O}(V_m)$.

The steps of the proposed adaptive sample size scheme is summarized in Algorithm 1. Note that since computation of the sub-optimality $R_n(\mathbf{w}_n) - R_n(\mathbf{w}_n^*)$ requires access to the minimizer $\mathbf{w}_n^*$, we replace the condition $R_n(\mathbf{w}_n) - R_n(\mathbf{w}_n^*) \leq V_n$ by a bound on the norm of gradient $\|\nabla R_n(\mathbf{w}_n)\|^2$. The risk $R_n$ is strongly convex, and we can bound the suboptimality $R_n(\mathbf{w}_n) - R_n(\mathbf{w}_n^*)$ as

$$R_n(\mathbf{w}_n) - R_n(\mathbf{w}_n^*) \leq \frac{1}{2cV_n}\|\nabla R_n(\mathbf{w}_n)\|^2. \quad (7)$$

Hence, at each stage, we stop updating the variable if the condition $\|\nabla R_n(\mathbf{w}_n)\| \leq (\sqrt{2c})V_n$ holds which implies $R_n(\mathbf{w}_n) - R_n(\mathbf{w}_n^*) \leq V_n$. The intermediate variable $\tilde{\mathbf{w}}$ can be updated in Step 7 using any first-order method. We will discuss this procedure for accelerated gradient descent (AGD) and stochastic variance reduced gradient (SVRG) methods in Sections 4.1 and 4.2, respectively.

# 4 Complexity Analysis

In this section, we aim to characterize the number of required iterations $s_n$ at each stage to solve the subproblems within their statistical accuracy. We derive this result for all linearly convergent first-order deterministic and stochastic methods.

The inequality in (6) not only leads to an efficient policy for the required precision $\delta_m$ at each step, but also provides an upper bound for the sub-optimality of the initial iterate, i.e., $\mathbf{w}_m$, for minimizing the risk $R_n$. Using this upper bound, depending on the iterative method of choice, we can characterize the number of required iterations $s_n$ to ensure that the updated variable is within the statistical accuracy of the risk $R_n$. To formally characterize the number of required iterations $s_n$, we first assume the following conditions are satisfied.

**Assumption 1.** *The loss functions $f(\mathbf{w}, \mathbf{z})$ are convex with respect to $\mathbf{w}$ for all values of $\mathbf{z}$. Moreover, their gradients $\nabla f(\mathbf{w}, \mathbf{z})$ are Lipschitz continuous with constant $M$*

$$\|\nabla f(\mathbf{w}, \mathbf{z}) - \nabla f(\mathbf{w}', \mathbf{z})\| \leq M \|\mathbf{w} - \mathbf{w}'\|, \qquad for\ all\ \mathbf{z}. \tag{8}$$

The conditions in Assumption 1 imply that the average loss $L(\mathbf{w})$ and the empirical loss $L_n(\mathbf{w})$ are convex and their gradients are Lipschitz continuous with constant $M$. Thus, the empirical risk $R_n(\mathbf{w})$ is strongly convex with constant $cV_n$ and its gradients $\nabla R_n(\mathbf{w})$ are Lipschitz continuous with parameter $M + cV_n$.

So far we have concluded that each subproblem should be solved up to its statistical accuracy. This observation leads to an upper bound for the number of iterations needed at each step to solve each subproblem. Indeed various descent methods can be executed for solving the sub-problem. Here we intend to come up with a general result that contains all descent methods that have a linear convergence rate when the objective function is strongly convex and smooth. In the following theorem, we derive a lower bound for the number of required iterations $s_n$ to ensure that the variable $\mathbf{w}_n$, which is the outcome of updating $\mathbf{w}_m$ by $s_n$ iterations of the method of interest, is within the statistical accuracy of the risk $R_n$ for any linearly convergent method.

**Theorem 2.** *Consider the variable $\mathbf{w}_m$ as a $V_m$-suboptimal solution of the risk $R_m$ in expectation, i.e., $\mathbb{E}[R_m(\mathbf{w}_m) - R_m(\mathbf{w}_m^*)] \leq V_m$, where $V_m = \mathcal{O}(1/m^\alpha)$. Consider the sets $\mathcal{S}_m \subset \mathcal{S}_n \subset \mathcal{T}$ such that $n = 2m$, and suppose Assumption 1 holds. Further, define $0 \leq \rho_n < 1$ as the linear convergence factor of the descent method used for updating the iterates. Then, the variable $\mathbf{w}_n$ generated based on the adaptive sample size mechanism satisfies $\mathbb{E}[R_n(\mathbf{w}_n) - R_n(\mathbf{w}_n^*)] \leq V_n$ if the number of iterations $s_n$ at the $n$-th stage is larger than*

$$s_n \geq -\frac{\log\left[3 \times 2^\alpha + (2^\alpha - 1)\left(2 + \frac{c}{2}\|\mathbf{w}^*\|^2\right)\right]}{\log \rho_n}. \tag{9}$$

*Proof.* See Section 7.2 in the supplementary material. □

The result in Theorem 2 characterizes the number of required iterations at each phase. Depending on the linear convergence factor $\rho_n$ and the parameter $\alpha$ for the order of statistical accuracy, the number of required iterations might be different. Note that the parameter $\rho_n$ might depend on the size of the training set directly or through the dependency of the problem condition number on $n$. It is worth mentioning that the result in (9) shows a lower bound for the number of required iteration which means that $s_n = \lfloor -(\log\left[3 \times 2^\alpha + (2^\alpha - 1)\left(2 + (c/2)\|\mathbf{w}^*\|^2\right)\right]/\log \rho_n)\rfloor + 1$ is the exact number of iterations needed when minimizing $R_n$, where $\lfloor a \rfloor$ indicates the floor of $a$. To characterize the overall computational complexity of the proposed adaptive sample size scheme, the exact expression for the linear convergence constant $\rho_n$ is required. In the following section, we focus on two deterministic and stochastic methods and characterize their overall computational complexity to reach the statistical accuracy of the full training set $\mathcal{T}$.

## 4.1 Adaptive Sample Size Accelerated Gradient (Ada AGD)

The accelerated gradient descent (AGD) method, also called as Nesterov's method, is a long-established descent method which achieves the optimal convergence rate for first-order deterministic methods. In this section, we aim to combine the update of AGD with the adaptive sample size scheme in Section 3 to improve convergence guarantees of AGD for solving ERM problems. This

can be done by using AGD for updating the iterates in step 7 of Algorithm 1. Given an iterate $\mathbf{w}_m$ within the statistical accuracy of the set $\mathcal{S}_m$, the adaptive sample size accelerated gradient descent method (Ada AGD) requires $s_n$ iterations of AGD to ensure that the resulted iterate $\mathbf{w}_n$ lies in the statistical accuracy of $\mathcal{S}_n$. In particular, if we initialize the sequences $\tilde{\mathbf{w}}$ and $\tilde{\mathbf{y}}$ as $\tilde{\mathbf{w}}_0 = \tilde{\mathbf{y}}_0 = \mathbf{w}_m$, the approximate solution $\mathbf{w}_n$ for the risk $R_n$ is the outcome of the updates

$$\tilde{\mathbf{w}}_{k+1} = \tilde{\mathbf{y}}_k - \eta_n \nabla R_n(\tilde{\mathbf{y}}_k), \tag{10}$$

and

$$\tilde{\mathbf{y}}_{k+1} = \tilde{\mathbf{w}}_{k+1} + \beta_n(\tilde{\mathbf{w}}_{k+1} - \tilde{\mathbf{w}}_k) \tag{11}$$

after $s_n$ iterations, i.e., $\mathbf{w}_n = \tilde{\mathbf{w}}_{s_n}$. The parameters $\eta_n$ and $\beta_n$ are indexed by $n$ since they depend on the number of samples. We use the convergence rate of AGD to characterize the number of required iterations $s_n$ to guarantee that the outcome of the recursive updates in (10) and (11) is within the statistical accuracy of $R_n$.

**Theorem 3.** *Consider the variable $\mathbf{w}_m$ as a $V_m$-optimal solution of the risk $R_m$ in expectation, i.e., $\mathbb{E}[R_m(\mathbf{w}_m) - R_m(\mathbf{w}_m^*)] \leq V_m$, where $V_m = \gamma/m^\alpha$. Consider the sets $\mathcal{S}_m \subset \mathcal{S}_n \subset \mathcal{T}$ such that $n = 2m$, and suppose Assumption 1 holds. Further, set the parameters $\eta_n$ and $\beta_n$ as*

$$\eta_n = \frac{1}{cV_n + M} \qquad and \qquad \beta_n = \frac{\sqrt{cV_n + M} - \sqrt{cV_n}}{\sqrt{cV_n + M} + \sqrt{cV_n}}. \tag{12}$$

*Then, the variable $\mathbf{w}_n$ generated based on the update of Ada AGD in (10)-(11) satisfies $\mathbb{E}[R_n(\mathbf{w}_n) - R_n(\mathbf{w}_n^*)] \leq V_n$ if the number of iterations $s_n$ is larger than*

$$s_n \geq \sqrt{\frac{n^\alpha M + c\gamma}{c\gamma}} \log \left[ 6 \times 2^\alpha + (2^\alpha - 1)\left(4 + c\|\mathbf{w}^*\|^2\right) \right]. \tag{13}$$

*Moreover, if we define $m_0$ as the size of the first training set, to reach the statistical accuracy $V_N$ of the full training set $\mathcal{T}$ the overall computational complexity of Ada GD is given by*

$$N\left[ 1 + \log_2\left(\frac{N}{m_0}\right) + \left(\frac{\sqrt{2^\alpha}}{\sqrt{2^\alpha} - 1}\right)\sqrt{\frac{N^\alpha M}{c\gamma}} \right] \log \left[ 6 \times 2^\alpha + (2^\alpha - 1)\left(4 + c\|\mathbf{w}^*\|^2\right) \right]. \tag{14}$$

*Proof.* See Section 7.3 in the supplementary material. □

The result in Theorem 3 characterizes the number of required iterations $s_n$ to achieve the statistical accuracy of $R_n$. Moreover, it shows that to reach the accuracy $V_N = \mathcal{O}(1/N^\alpha)$ for the risk $R_N$ accosiated to the full training set $\mathcal{T}$, the total computational complexity of Ada AGD is of the order $\mathcal{O}\left(N^{(1+\alpha/2)}\right)$. Indeed, this complexity is lower than the overall computational complexity of AGD for reaching the same target which is given by $\mathcal{O}\left(N\sqrt{\kappa_N}\log(N^\alpha)\right) = \mathcal{O}\left(N^{(1+\alpha/2)}\log(N^\alpha)\right)$. Note that this bound holds for AGD since the condition number $\kappa_N := (M + cV_N)/(cV_N)$ of the risk $R_N$ is of the order $\mathcal{O}(1/V_N) = \mathcal{O}(N^\alpha)$.

## 4.2 Adaptive Sample Size SVRG (Ada SVRG)

For the adaptive sample size mechanism presented in Section 3, we can also use linearly convergent *stochastic* methods such as stochastic variance reduced gradient (SVRG) in [10] to update the iterates. The SVRG method succeeds in reducing the computational complexity of deterministic first-order methods by computing a single gradient per iteration and using a *delayed* version of the average gradient to update the iterates. Indeed, we can exploit the idea of SVRG to develop low computational complexity adaptive sample size methods to improve the performance of deterministic adaptive sample size algorithms. Moreover, the adaptive sample size variant of SVRG (Ada SVRG) enhances the proven bounds for SVRG to solve ERM problems.

We proceed to extend the idea of adaptive sample size scheme to the SVRG algorithm. To do so, consider $\mathbf{w}_m$ as an iterate within the statistical accuracy, $\mathbb{E}[R_m(\mathbf{w}_m) - R_m(\mathbf{w}_m^*)] \leq V_m$, for a set $\mathcal{S}_m$ which contains $m$ samples. Consider $s_n$ and $q_n$ as the numbers of outer and inner loops for the update of SVRG, respectively, when the size of the training set is $n$. Further, consider $\tilde{\mathbf{w}}$ and $\hat{\mathbf{w}}$ as the sequences of iterates for the outer and inner loops of SVRG, respectively. In the adaptive sample

size SVRG (Ada SVRG) method to minimize the risk $R_n$, we set the approximate solution $\mathbf{w}_m$ for the previous ERM problem as the initial iterate for the outer loop, i.e., $\tilde{\mathbf{w}}_0 = \mathbf{w}_m$. Then, the outer loop update which contains gradient computation is defined as

$$\nabla R_n(\tilde{\mathbf{w}}_k) = \frac{1}{n} \sum_{i=1}^{n} \nabla f(\tilde{\mathbf{w}}_k, z_i) + cV_n \tilde{\mathbf{w}}_k \qquad \text{for} \qquad k = 0, \ldots, s_n - 1, \qquad (15)$$

and the inner loop for the $k$-th outer loop contains $q_n$ iterations of the following update

$$\hat{\mathbf{w}}_{t+1,k} = \hat{\mathbf{w}}_{t,k} - \eta_n \left( \nabla f(\hat{\mathbf{w}}_{t,k}, z_{i_t}) + cV_n \hat{\mathbf{w}}_{t,k} - \nabla f(\tilde{\mathbf{w}}_k, z_{i_t}) - cV_n \tilde{\mathbf{w}}_k + \nabla R_n(\tilde{\mathbf{w}}_k) \right), \quad (16)$$

for $t = 0, \ldots, q_n - 1$, where the iterates for the inner loop at step $k$ are initialized as $\hat{\mathbf{w}}_{0,k} = \tilde{\mathbf{w}}_k$, and $i_t$ is index of the function which is chosen unfirmly at random from the set $\{1, \ldots, n\}$ at the inner iterate $t$. The outcome of each inner loop $\hat{\mathbf{w}}_{q_n,k}$ is used as the variable for the next outer loop, i.e., $\tilde{\mathbf{w}}_{k+1} = \hat{\mathbf{w}}_{q_n,k}$. We define the outcome of $s_n$ outer loops $\tilde{\mathbf{w}}_{s_n}$ as the approximate solution for the risk $R_n$, i.e., $\mathbf{w}_n = \tilde{\mathbf{w}}_{s_n}$.

In the following theorem we derive a bound on the number of required outer loops $s_n$ to ensure that the variable $\mathbf{w}_n$ generated by the updates in (15) and (16) will be in the statistical accuracy of $R_n$ in expectation, i.e., $\mathbb{E}[R_n(\mathbf{w}_n) - R_n(\mathbf{w}_n^*)] \leq V_n$. To reach the smallest possible lower bound for $s_n$, we properly choose the number of inner loop iterations $q_n$ and the learning rate $\eta_n$.

**Theorem 4.** *Consider the variable $\mathbf{w}_m$ as a $V_m$-optimal solution of the risk $R_m$, i.e., a solution such that $\mathbb{E}[R_m(\mathbf{w}_m) - R_m(\mathbf{w}_m^*)] \leq V_m$, where $V_m = \mathcal{O}(1/m^\alpha)$. Consider the sets $\mathcal{S}_m \subset \mathcal{S}_n \subset \mathcal{T}$ such that $n = 2m$, and suppose Assumption 1 holds. Further, set the number of inner loop iterations as $q_n = n$ and the learning rate as $\eta_n = 0.1/(M + cV_n)$. Then, the variable $\mathbf{w}_n$ generated based on the update of Ada SVRG in (15)-(16) satisfies $\mathbb{E}[R_n(\mathbf{w}_n) - R_n(\mathbf{w}_n^*)] \leq V_n$ if the number of iterations $s_n$ is larger than*

$$s_n \geq \log_2 \left[ 3 \times 2^\alpha + (2^\alpha - 1) \left( 2 + \frac{c}{2} \|\mathbf{w}^*\|^2 \right) \right]. \qquad (17)$$

*Moreover, to reach the statistical accuracy $V_N$ of the full training set $\mathcal{T}$ the overall computational complexity of Ada SVRG is given by*

$$4N \, \log_2 \left[ 3 \times 2^\alpha + (2^\alpha - 1) \left( 2 + \frac{c}{2} \|\mathbf{w}^*\|^2 \right) \right]. \qquad (18)$$

*Proof.* See Section 7.4. □

The result in (17) shows that the minimum number of outer loop iterations for Ada SVRG is equal to $s_n = \lfloor \log_2[3 \times 2^\alpha + (2^\alpha - 1)(2 + (c/2)\|\mathbf{w}^*\|^2)] \rfloor + 1$. This bound leads to the result in (18) which shows that the overall computational complexity of Ada SVRG to reach the statistical accuracy of the full training set $\mathcal{T}$ is of the order $\mathcal{O}(N)$. This bound not only improves the bound $\mathcal{O}(N^{1+\alpha/2})$ for Ada AGD, but also enhances the complexity of SVRG for reaching the same target accuracy which is given by $\mathcal{O}((N + \kappa) \log(N^\alpha)) = \mathcal{O}(N \log(N^\alpha))$.

## 5 Experiments

In this section, we compare the adaptive sample size versions of a group of first-order methods, including gradient descent (GD), accelerated gradient descent (AGD), and stochastic variance reduced gradient (SVRG) with their standard (fixed sample size) versions. In the main paper, we only use the RCV1 dataset. Further numerical experiments on MNIST dataset can be found in Section 7.5 in the supplementary material. We use $N = 10,000$ samples of the RCV1 dataset for the training set and the remaining $10,242$ as the test set. The number of features in each sample is $p = 47,236$. In our experiments, we use logistic loss. The constant $c$ should be within the order of gradients Lipschitz continuity constant $M$, and, therefore, we set it as $c = 1$ since the samples are normalized and $M = 1$. The size of the initial training set for adaptive methods is $m_0 = 400$. In our experiments we assume $\alpha = 0.5$ and therefore the added regularization term is $(1/\sqrt{n})\|\mathbf{w}\|^2$.

The plots in Figure 1 compare the suboptimality of GD, AGD, and SVRG with their adaptive sample size versions. As our theoretical results suggested, we observe that the adaptive sample size scheme reduces the overall computational complexity of all of the considered linearly convergent first-order

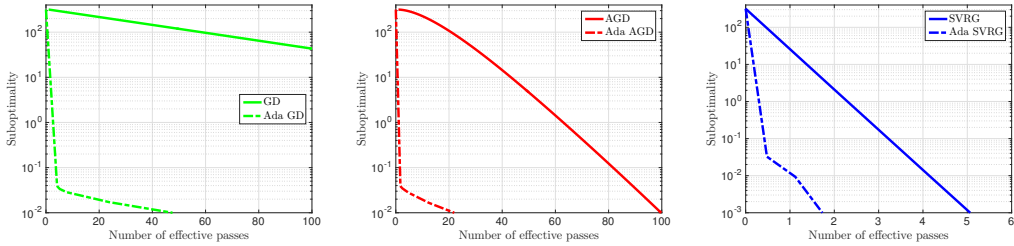

Figure 1: Suboptimality vs. number of effective passes for RCV1 dataset with regularization of $\mathcal{O}(1/\sqrt{n})$.

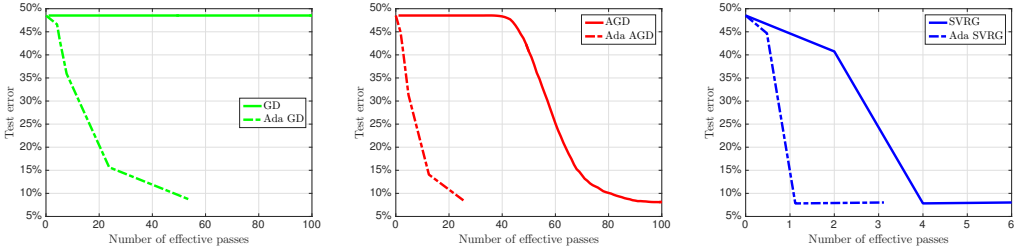

Figure 2: Test error vs. number of effective passes for RCV1 dataset with regularization of $\mathcal{O}(1/\sqrt{n})$.

methods. If we compare the test errors of GD, AGD, and SVRG with their adaptive sample size variants, we reach the same conclusion that the adaptive sample size scheme reduces the overall computational complexity to reach the statistical accuracy of the full training set. In particular, the left plot in Figure 2 shows that Ada GD approaches the minimum test error of $8\%$ after $55$ effective passes, while GD can not improve the test error even after $100$ passes. Indeed, GD will reach lower test error if we run it for more iterations. The central plot in Figure 2 showcases that Ada AGD reaches $8\%$ test error about $5$ times faster than AGD. This is as predicted by $\log(N^\alpha) = \log(100) = 4.6$. The right plot in Figure 2 illustrates a similar improvement for Ada SVRG. We have observed similar performances for other datasets such as MNIST – see Section 7.5 in supplementary material.

## 6 Discussions

We presented an adaptive sample size scheme to improve the convergence guarantees for a class of first-order methods which have linear convergence rates under strong convexity and smoothness assumptions. The logic behind the proposed adaptive sample size scheme is to replace the solution of a relatively *hard* problem – the ERM problem for the full training set – by a sequence of relatively *easier* problems – ERM problems corresponding to a subset of samples. Indeed, whenever $m < n$, solving the ERM problems in (4) for loss $R_m$ is simpler than the one for loss $R_n$ because:

  (i) The adaptive regularization term of order $V_m$ makes the condition number of $R_m$ smaller than the condition number of $R_n$ – which uses a regularizer of order $V_n$.

 (ii) The approximate solution $\mathbf{w}_m$ that we need to find for $R_m$ is less accurate than the approximate solution $\mathbf{w}_n$ we need to find for $R_n$.

(iii) The computation cost of an iteration for $R_m$ – e.g., the cost of evaluating a gradient – is lower than the cost of an iteration for $R_n$.

Properties (i)-(iii) combined with the ability to grow the sample size geometrically, reduce the overall computational complexity for reaching the statistical accuracy of the full training set. We particularized our results to develop adaptive (Ada) versions of AGD and SVRG. In both methods we found a computational complexity reduction of order $\mathcal{O}(\log(1/V_N)) = \mathcal{O}(\log(N^\alpha))$ which was corroborated in numerical experiments. The idea and analysis of adaptive first order methods apply generically to any other approach with linear convergence rate (Theorem 2). The development of sample size adaptation for sublinear methods is left for future research.

### Acknowledgments

This research was supported by NSF CCF 1717120 and ARO W911NF1710438.

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
