[Supplementary Material]

# 7 Supplementary Material

## 7.1 Proof of Proposition 1

The steps of the proof for Proposition 1 are adopted from the analysis in [12]. We start the proof by providing an upper bound for the difference between the loss functions $L_n$ and $L_m$. The upper bound is studied in the following lemma which uses the condition in (3).

**Lemma 5.** *Consider $L_n$ and $L_m$ as the empirical losses of the sets $\mathcal{S}_n$ and $\mathcal{S}_m$, respectively, where they are chosen such that $\mathcal{S}_m \subset \mathcal{S}_n$. If we define $n$ and $m$ as the number of samples in the training sets $\mathcal{S}_n$ and $\mathcal{S}_m$, respectively, then the expected absolute value of the difference between the empirical losses is bounded above by*

$$\mathbb{E}\left[|L_n(\mathbf{w}) - L_m(\mathbf{w})|\right] \leq \frac{n-m}{n}\left(V_{n-m} + V_m\right), \tag{19}$$

*for any $\mathbf{w}$.*

*Proof.* First we characterize the difference between the difference of the loss functions associated with the sets $\mathcal{S}_m$ and $\mathcal{S}_n$. To do so, consider the difference

$$L_n(\mathbf{w}) - L_m(\mathbf{w}) = \frac{1}{n}\sum_{i \in \mathcal{S}_n} f_i(\mathbf{w}) - \frac{1}{m}\sum_{i \in \mathcal{S}_m} f_i(\mathbf{w}). \tag{20}$$

Notice that the set $\mathcal{S}_m$ is a subset of the set $\mathcal{S}_n$ and we can write $\mathcal{S}_n = \mathcal{S}_m \cup \mathcal{S}_{n-m}$. Thus, we can rewrite the right hand side of (20) as

$$L_n(\mathbf{w}) - L_m(\mathbf{w}) = \frac{1}{n}\left[\sum_{i \in \mathcal{S}_m} f_i(\mathbf{w}) + \sum_{i \in \mathcal{S}_{n-m}} f_i(\mathbf{w})\right] - \frac{1}{m}\sum_{i \in \mathcal{S}_m} f_i(\mathbf{w})$$

$$= \frac{1}{n}\sum_{i \in \mathcal{S}_{n-m}} f_i(\mathbf{w}) - \frac{n-m}{mn}\sum_{i \in \mathcal{S}_m} f_i(\mathbf{w}). \tag{21}$$

Factoring $(n-m)/n$ from the terms in the right hand side of (21) follows

$$L_n(\mathbf{w}) - L_m(\mathbf{w}) = \frac{n-m}{n}\left[\frac{1}{n-m}\sum_{i \in \mathcal{S}_{n-m}} f_i(\mathbf{w}) - \frac{1}{m}\sum_{i \in \mathcal{S}_m} f_i(\mathbf{w})\right]. \tag{22}$$

Now add and subtract the statistical loss $L(\mathbf{w})$ and compute the expected value to obtain

$$\mathbb{E}[|L_n(\mathbf{w}) - L_m(\mathbf{w})|] = \frac{n-m}{n}\mathbb{E}\left[\left|\frac{1}{n-m}\sum_{i \in \mathcal{S}_{n-m}} f_i(\mathbf{w}) - L(\mathbf{w}) + L(\mathbf{w}) - \frac{1}{m}\sum_{i \in \mathcal{S}_m} f_i(\mathbf{w})\right|\right]$$

$$\leq \frac{n-m}{n}\left(V_{n-m} + V_m\right), \tag{23}$$

where the last inequality follows by using the triangle inequality and the upper bound in (3). $\square$

The result in Lemma 5 shows that the upper bound for the difference between the loss functions associated with the sets $\mathcal{S}_m$ and $\mathcal{S}_n$ where $\mathcal{S}_m \subset \mathcal{S}_n$ is proportional to the difference between the size of these two sets $n-m$.

In the following lemma, we characterize an upper bound for the norm of the optimal argument $\mathbf{w}_n^*$ of the empirical risk $R_n(\mathbf{w})$ in terms of the norm of statistical average loss $L(\mathbf{w})$ optimal argument $\mathbf{w}^*$.

**Lemma 6.** *Consider $L_n$ as the empirical loss of the set $\mathcal{S}_n$ and $L$ as the statistical average loss. Moreover, recall $\mathbf{w}^*$ as the optimal argument of the statistical average loss $L$, i.e., $\mathbf{w}^* = argmin_{\mathbf{w}} L(\mathbf{w})$. If Assumption 1 holds, then the norm of the optimal argument $\mathbf{w}_n^*$ of the regularized empirical risk $R_n(\mathbf{w}) := L_n(\mathbf{w}) + cV_n\|\mathbf{w}\|^2$ is bounded above by*

$$\mathbb{E}[\|\mathbf{w}_n^*\|^2] \leq \frac{4}{c} + \|\mathbf{w}^*\|^2 \tag{24}$$

*Proof.* The optimality condition of $\mathbf{w}_n^*$ for the the regularized empirical risk $R_n(\mathbf{w}) = L_n(\mathbf{w}) + (cV_n)/2\|\mathbf{w}\|^2$ implies that

$$L_n(\mathbf{w}_n^*) + \frac{cV_n}{2}\|\mathbf{w}_n^*\|^2 \le L_n(\mathbf{w}^*) + \frac{cV_n}{2}\|\mathbf{w}^*\|^2. \tag{25}$$

By regrouping the terms and computing the expectation we can show that $\mathbb{E}[\|\mathbf{w}_n^*\|^2]$ is bounded above by

$$\mathbb{E}[\|\mathbf{w}_n^*\|^2] \le \frac{2}{cV_n}\mathbb{E}[(L_n(\mathbf{w}^*) - L_n(\mathbf{w}_n^*))] + \|\mathbf{w}^*\|^2. \tag{26}$$

We proceed to bound the difference $L_n(\mathbf{w}^*) - L_n(\mathbf{w}_n^*)$. By adding and subtracting the terms $L(\mathbf{w}^*)$ and $L(\mathbf{w}_n^*)$ we obtain that

$$L_n(\mathbf{w}^*) - L_n(\mathbf{w}_n^*) = \left[L_n(\mathbf{w}^*) - L(\mathbf{w}^*)\right] + \left[L(\mathbf{w}^*) - L(\mathbf{w}_n^*)\right] + \left[L(\mathbf{w}_n^*) - L_n(\mathbf{w}_n^*)\right]. \tag{27}$$

Notice that the second bracket in (27) is non-positive since $L(\mathbf{w}^*) \le L(\mathbf{w}_n^*)$. Therefore, it is bounded by 0. According to (3), the first and third brackets in (27) are bounded above by $V_n$ in expectation. Replacing these upper bounds by the brackets in (27) yields

$$\mathbb{E}[L_n(\mathbf{w}^*) - L_n(\mathbf{w}_n^*)] \le 2V_n. \tag{28}$$

Substituting the upper bound in (28) into (26) implies the claim in (24). $\quad\square$

Note that the difference $R_n(\mathbf{w}_m) - R_n(\mathbf{w}_n^*)$ can be written as

$$\begin{aligned} R_n(\mathbf{w}_m) - R_n(\mathbf{w}_n^*) = R_n(\mathbf{w}_m) - R_m(\mathbf{w}_m) + R_m(\mathbf{w}_m) - R_m(\mathbf{w}_m^*) \\ + R_m(\mathbf{w}_m^*) - R_m(\mathbf{w}_n^*) + R_m(\mathbf{w}_n^*) - R_n(\mathbf{w}_n^*). \end{aligned} \tag{29}$$

We proceed to bound the differences in (29). To do so, note that the difference $R_n(\mathbf{w}_m) - R_m(\mathbf{w}_m)$ can be simplified as

$$\begin{aligned} R_n(\mathbf{w}_m) - R_m(\mathbf{w}_m) &= L_n(\mathbf{w}_m) - L_m(\mathbf{w}_m) + \frac{c(V_n - V_m)}{2}\|\mathbf{w}_m\|^2 \\ &\le L_n(\mathbf{w}) - L_m(\mathbf{w}), \end{aligned} \tag{30}$$

where the inequality follows from the fact that $V_n < V_m$ and $V_n - V_m$ is negative. It follows from the result in Lemma 5 that the right hand side of (30) is bounded by $(n-m)/n\,(V_{n-m} + V_m)$. Therefore,

$$\mathbb{E}\left[|R_n(\mathbf{w}_m) - R_m(\mathbf{w}_m)|\right] \le \frac{n-m}{n}\left(V_{n-m} + V_m\right). \tag{31}$$

According to the fact that $\mathbf{w}_m$ as an $\delta_m$ optimal solution for the sub-optimality $\mathbb{E}\left[R_m(\mathbf{w}_m) - R_m(\mathbf{w}_m^*)\right]$ we know that

$$\mathbb{E}[R_m(\mathbf{w}_m) - R_m(\mathbf{w}_m^*)] \le \delta_m. \tag{32}$$

Based on the definition of $\mathbf{w}_m^*$ which is the optimal solution of the risk $R_m$, the third difference in (29) which is $R_m(\mathbf{w}_m^*) - R_m(\mathbf{w}_n^*)$ is always negative. I.e.,

$$R_m(\mathbf{w}_m^*) - R_m(\mathbf{w}_n^*) \le 0. \tag{33}$$

Moreover, we can use the triangle inequality to bound the difference $R_m(\mathbf{w}_n^*) - R_n(\mathbf{w}_n^*)$ in (29) as

$$\begin{aligned} \mathbb{E}[R_m(\mathbf{w}_n^*) - R_n(\mathbf{w}_n^*)] &= \mathbb{E}[L_m(\mathbf{w}_n^*) - L_n(\mathbf{w}_n^*)] + \frac{c(V_m - V_n)}{2}\mathbb{E}[\|\mathbf{w}_n^*\|^2] \\ &\le \frac{n-m}{n}\left(V_{n-m} + V_m\right) + \frac{c(V_m - V_n)}{2}\mathbb{E}[\|\mathbf{w}_n^*\|^2]. \end{aligned} \tag{34}$$

Replacing the differences in (29) by the upper bounds in (31)-(34) leads to

$$\mathbb{E}[R_n(\mathbf{w}_m) - R_n(\mathbf{w}_n^*)] \le \delta_m + \frac{2(n-m)}{n}\left(V_{n-m} + V_m\right) + \frac{c(V_m - V_n)}{2}\mathbb{E}[\|\mathbf{w}_n^*\|^2] \tag{35}$$

Substitute $\mathbb{E}[\|\mathbf{w}_n^*\|^2]$ in (35) by the upper bound in (24) to obtain the result in (5).

## 7.2 Proof of Theorem 2

According to the result in Proposition 1 and the condition that $\mathbb{E}[R_m(\mathbf{w}_m) - R_m(\mathbf{w}_m^*)] \leq V_m$, we obtain that

$$\mathbb{E}[R_n(\mathbf{w}_m) - R_n(\mathbf{w}_n^*)] \leq \left[3 + \left(1 - \frac{1}{2^\alpha}\right)\left(2 + \frac{c}{2}\|\mathbf{w}^*\|^2\right)\right] V_m. \tag{36}$$

If we assume that the first-order descent method that we use to update the iterates has a linear convergence rate, then there exists a constant $0 < \rho_n < 1$ we obtain that after $s_n$ iterations the error is bounded above by

$$R_n(\mathbf{w}_n) - R_n(\mathbf{w}_n^*) \leq \rho_n^{s_n}(R_n(\mathbf{w}_m) - R_n(\mathbf{w}_n^*)). \tag{37}$$

The result in (37) holds for deterministic methods. If we use a stochastic linearly convergent method such as SVRG, then the result holds in expectation and we can write

$$\mathbb{E}[R_n(\mathbf{w}_n) - R_n(\mathbf{w}_n^*)] \leq \rho^{s_n}(R_n(\mathbf{w}_m) - R_n(\mathbf{w}_n^*)), \tag{38}$$

where the expectation is with respect to the index of randomly chosen functions.

It follows form computing the expected value of both sides in (37) with respect to the choice of training sets and using the upper bound in (36) for the expected difference $\mathbb{E}[R_n(\mathbf{w}_m) - R_n(\mathbf{w}_n^*)]$ that

$$\mathbb{E}[R_n(\mathbf{w}_n) - R_n(\mathbf{w}_n^*)] \leq \rho^{s_n}\left[3 + \left(1 - \frac{1}{2^\alpha}\right)\left(2 + \frac{c}{2}\|\mathbf{w}^*\|^2\right)\right] V_m. \tag{39}$$

Note that the inequality in (39) also holds for stochastic methods. The difference is in stochastic methods the expectation is with respect to the choice of training sets and the index of random functions, while for deterministic methods it is only with respect to the choice of training sets.

To ensure that the suboptimality $\mathbb{E}[R_n(\mathbf{w}_n) - R_n(\mathbf{w}_n^*)]$ is smaller than $V_n$ we need to guarantee that the right hand side in (39) is not larger than $V_n$, which is equivalent to the condition

$$\rho^{s_n}\left[3 + \left(1 - \frac{1}{2^\alpha}\right)\left(2 + \frac{c}{2}\|\mathbf{w}^*\|^2\right)\right] \leq \frac{1}{2^\alpha}. \tag{40}$$

By regrouping the terms in (40) we obtain that

$$s_n \geq -\frac{\log\left[3 \times 2^\alpha + (2^\alpha - 1)\left(2 + \frac{c}{2}\|\mathbf{w}^*\|^2\right)\right]}{\log(\rho_n)}, \tag{41}$$

and the claim in (9) follows.

## 7.3 Proof of Theorem 3

Note that according to the convergence result for accelerated gradient descent in [13], the suboptimality of accelerated gradient descent method is linearly convergent with the constant $1 - 1/\sqrt{\kappa}$ where $\kappa$ is the condition number of the objective function. In particular, the suboptimality after $s_n$ iterations is bounded above by

$$R_n(\mathbf{w}_n) - R_n(\mathbf{w}_n^*) \leq \left(1 - \sqrt{\frac{1}{\kappa}}\right)^{s_n}\left(R_n(\mathbf{w}_m) - R_n(\mathbf{w}_n^*) + \frac{m}{2}\|\mathbf{w}_m - \mathbf{w}_n^*\|^2\right), \tag{42}$$

where $m$ is the constant of strong convexity. Replacing $\frac{m}{2}\|\mathbf{w}_m - \mathbf{w}_n^*\|^2$ by its upper bound $R_n(\mathbf{w}_m) - R_n(\mathbf{w}_n^*)$ leads to the expression

$$R_n(\mathbf{w}_n) - R_n(\mathbf{w}_n^*) \leq 2\left(1 - \sqrt{\frac{1}{\kappa}}\right)^{s_n}(R_n(\mathbf{w}_m) - R_n(\mathbf{w}_n^*)), \tag{43}$$

Hence, if we follow the steps of the proof of Theorem 2 we obtain that $s_n$ should be larger than

$$s_n \geq -\frac{\log\left[6 \times 2^\alpha + (2^\alpha - 1)\left(4 + c\|\mathbf{w}^*\|^2\right)\right]}{\log(1 - 1/\sqrt{\kappa})}. \tag{44}$$

According to the inequality $-\log(1-x) > x$, we can replace $-\log(1-1/\sqrt{\kappa})$ by its lower bound $1/\sqrt{\kappa}$ to obtain

$$s_n \geq \sqrt{\kappa_n} \log\left[6 \times 2^\alpha + (2^\alpha - 1)\left(4 + c\|\mathbf{w}^*\|^2\right)\right]. \tag{45}$$

Note if the condition in (45) holds, then the inequality in (44) follows. The condition number of the risk $R_n$ is given by $\kappa_n = (M + cV_n)/cV_n$. Further, as stated in the statement of the theorem, $V_n$ can be written as $V_n = \gamma/n^\alpha$ where $\gamma$ is a positive constant and $\alpha \in [0.5, 1]$. Based on these expressions, we can rewrite (45) as

$$s_n \geq \sqrt{\frac{n^\alpha M + c\gamma}{c\gamma}} \log\left[6 \times 2^\alpha + (2^\alpha - 1)\left(4 + c\|\mathbf{w}^*\|^2\right)\right], \tag{46}$$

which follows the claim in (13). If we assume that we start with $m_0$ samples such that $N/m_0 = 2^q$ where $q$ is an integer then the total number of gradient computations to achieve $V_N$ for the risk $R_N$ is given by

$$\sum_{n=m_0, 2m_0, \ldots, N} \sqrt{\frac{n^\alpha M + c\gamma}{c\gamma}} \log\left[6 \times 2^\alpha + (2^\alpha - 1)\left(4 + c\|\mathbf{w}^*\|^2\right)\right]$$

$$\leq \log\left[6 \times 2^\alpha + (2^\alpha - 1)\left(4 + c\|\mathbf{w}^*\|^2\right)\right] \sum_{n=m_0, 2m_0, \ldots, N} 1 + \sqrt{\frac{n^\alpha M}{c\gamma}}$$

$$= \log\left[6 \times 2^\alpha + (2^\alpha - 1)\left(4 + c\|\mathbf{w}^*\|^2\right)\right] \left[(q+1) + \sqrt{\frac{m_0^\alpha M}{c\gamma}} \left(\frac{\sqrt{2^{(q+1)\alpha}} - 1}{\sqrt{2^\alpha} - 1}\right)\right]$$

$$\leq \log\left[6 \times 2^\alpha + (2^\alpha - 1)\left(4 + c\|\mathbf{w}^*\|^2\right)\right] \left[(q+1) + \sqrt{\frac{m_0^\alpha M}{c\gamma}} \left(\frac{\sqrt{2^{(q+1)\alpha}}}{\sqrt{2^\alpha} - 1}\right)\right]$$

$$= \log\left[6 \times 2^\alpha + (2^\alpha - 1)\left(4 + c\|\mathbf{w}^*\|^2\right)\right] \left[(q+1) + \sqrt{\frac{N^\alpha M}{c\gamma}} \left(\frac{\sqrt{2^\alpha}}{\sqrt{2^\alpha} - 1}\right)\right]. \tag{47}$$

Replacing $q$ by $\log_2(N/m_0)$ leads to the bound in (14).

### 7.4 Proof of Theorem 4

Let's recall the convergence result of SVRG after $s$ outer loop where each inner loop contains $r$ iterations. We can show that if $\mathbf{w}_m$ is the variable corresponding to $m$ samples and $n$ is the variable associated with $n$ samples, then we have

$$\mathbf{E}_n[R_n(\mathbf{w}_n) - R_n(\mathbf{w}_n^*)] \leq \rho^s \left[R_n(\mathbf{w}_m) - R_n(\mathbf{w}_n^*)\right], \tag{48}$$

where the expectation is taken with respect to the indices chosen in the inner loops, and the constant $\rho$ is defined as

$$\rho := \frac{1}{\gamma\eta(1 - 2L_0\eta)r} + \frac{2L_0\eta}{1 - 2L_0\eta} < 1 \tag{49}$$

where $\gamma$ is the constant of strong convexity, $L_0$ is the constant for the Lipschitz continuity of gradients, $q$ is the number of inner loop iterations, and $\eta$ is the stepsize. If we assume that $V_n = \mathcal{O}(1/n^\alpha)$, then we obtain that $\gamma = c/n^\alpha$ and $L_0 = M + c/n^\alpha$. Further, if we set the number of inner loop iteration as $q = n$ and the stepsize as $\eta = 0.1/L_0$, the expression for $\rho$ can be simplified as

$$\rho := \frac{Mn^\alpha + c}{0.08nc} + \frac{1}{4} < \frac{1}{2}, \tag{50}$$

where the inequality holds since the size of training set is such that $(Mn^\alpha + c)/(nc) \leq 0.02$. Considering the result in (41) and the upper bound for the linear factor $\rho$, to ensure that that outocme of the Ada SVRG is within the statistical accuracy of the risk $R_n$ the number of outer loops $s_n$ should be larger than

$$s_n \geq \log_2\left[3 \times 2^\alpha + (2^\alpha - 1)\left(2 + \frac{c}{2}\|\mathbf{w}^*\|^2\right)\right], \tag{51}$$

and the result in (17) follows.

Since each outer loop requires one full gradient computation and $n$ inner loop iterations the total number of gradient computations (computational complexity) of Ada SVRG at the stage of minimizing $R_n$ is given by $2ns_n$. Therefore, if we assume that we start with $m_0$ samples such that $N/m_0 = 2^q$ where $q$ is an integer, then the total number of gradient computations to achieve $V_N$ for the risk $R_N$ is given by

$$
\sum_{n=m_0, 2m_0, \ldots, N} 2n \log_2 \left[ 3 \times 2^\alpha + (2^\alpha - 1) \left( 2 + \frac{c}{2} \|\mathbf{w}^*\|^2 \right) \right]
$$
$$
= 2m_0 \frac{2^{q+1} - 1}{2 - 1} \log_2 \left[ 3 \times 2^\alpha + (2^\alpha - 1) \left( 2 + \frac{c}{2} \|\mathbf{w}^*\|^2 \right) \right]
$$
$$
\leq 4N \log_2 \left[ 3 \times 2^\alpha + (2^\alpha - 1) \left( 2 + \frac{c}{2} \|\mathbf{w}^*\|^2 \right) \right], \tag{52}
$$

which yields the claim in (18).

### 7.5   Additional Numerical Experiments

For the experiments of this section we use the MNIST dataset containing images of dimension $p = 784$. Since we are interested in a binary classification problem we only use the samples corresponding to digits $0$ and $8$, and, therefore, the number of samples is $11,774$. We choose $N = 6,000$ of these samples randomly and use them as the training set and use the remaining $5,774$ samples as the test set. We use the logistic loss to evaluate the performance of the classifier and normalize the samples to ensure that the constant for the Lipschitz continuity of the gradients is $M = 1$. In our experiments we consider two different scenarios. First we compare GD, AGD, and SVRG with their adaptive sample size versions when the additive regularization term is of order $1/\sqrt{n}$. Then, we redo the experiments for a regularization term of order $1/n$.

The plots in Figure 3 compare the suboptimality of GD, AGD, and SVRG with Ada GD, Ada AGD, and Ada SVRG when the regularization term in $(1/\sqrt{n})\|\mathbf{w}\|^2$. Note that in this case the statistical accuracy should be order of $\mathcal{O}(1/\sqrt{n})$ and therefore we are interested in the number of required iterations to achieve the suboptimality of order $10^{-2}$. As we observe Ada GD reach this target accuracy almost 6 times faster than GD. The improvement for Ada AGD and Ada SVRG is less significant, but they still reach the suboptimality of $10^{-2}$ significantly faster than their standard (fixed sample size) methods. Figure 4 illustrates the test error of GD, AGD, SVRG, Ada GD, Ada AGD, and Ada SVRG versus the number of effective passes over the dataset when the added regularization is of the order $\mathcal{O}(1/\sqrt{n})$. Comparison of these methods in terms of test error also support the gain in solving subproblems sequentially instead of minimizing the ERM corresponding to the full training set directly. In particular, for all three methods, the adaptive sample size version reaches the minimum test error of $2.5\%$ faster than the fixed sample size version.

We also run the same experiments for the case that the regularization term is order $1/n$. Figure 5 shows the suboptimality of GD, AGD, and SVRG and their adaptive sample size version for the MNIST dataset when $V_n$ is assumed to be $\mathcal{O}(1/n)$. We expect from our theoretical achievements the advantage of using adaptive sample size scheme in this setting should be more significant, since $\log(N)$ is twice the value of $\log(\sqrt{N})$. Figure 5 fulfills this expectation by showing that Ada GD, Ada AGD, and Ada SVRG are almost 10 times faster than GD, AGD, and SVRG, respectively. Figure 6 demonstrates the test error of these methods versus the number of effective passes for a regularization of order $\mathcal{O}(1/n)$. In this case, this case all methods require more passes to achieve the minimum test error comparing to the case that regularization is of order $\mathcal{O}(1/n)$. Interestingly, the minimum accuracy in this case is equal to $1\%$ which is lower than $2.5\%$ for the previous setting. Indeed, the difference between the number of required passes to reach the minimum test error for adaptive sample size methods and their standard version is more significant since the factor $\log(N^\alpha)$ is larger.

Figure 3: Comparison of GD, AGD, and SVRG with their adaptive sample size versions in terms of suboptimality vs. number of effective passes for MNIST dataset with regularization of the order $\mathcal{O}(1/\sqrt{n})$.

Figure 4: Comparison of GD, AGD, and SVRG with their adaptive sample size versions in terms of test error vs. number of effective passes for MNIST dataset with regularization of the order $\mathcal{O}(1/\sqrt{n})$.

Figure 5: Comparison of GD, AGD, and SVRG with their adaptive sample size versions in terms of suboptimality vs. number of effective passes for MNIST dataset with regularization of the order $\mathcal{O}(1/n)$.

Figure 6: Comparison of GD, AGD, and SVRG with their adaptive sample size versions in terms of test error vs. number of effective passes for MNIST dataset with regularization of the order $\mathcal{O}(1/n)$.