[Reviews · NeurIPS 2017]

Reviewer 1



This paper presents algorithms for empirical risk minimization on large datasets using an adaptive sample size to improve bounds on convergence. The work is similar in spirit to DynaSAGA and AdaNewton, although the emphasis is on first order techniques instead of second order ones presented in Ada Newton. The problem formulation also uses a regularization term. A theoretical bound is provided for an acceptable accuracy between the empirical and statistical solution in Proposition 1. Furthermore, the required number of iterations s_n is lower bounded. The empirical results present a comparison of gradient descent, accelerated gradient descent and stochastic variance reduction techniques using the adaptive sample size technique. On RCV1 and MNIST datasets, it is seen that the overall computational cost to reach the optimal solution is significantly reduced. How does the choice of the learning rate affect s_n i.e. the number of iterations required to obtain convergence? No description of the execution environment is presented - is the code reusable? Minor comments: 1. Supplementary Material, Pg 2, Ln 397: bonded -> bounded 2. There appears to be no theorem 1.

Reviewer 2



The paper proposes an adaptive sample size strategy applied to first order methods to reduce the complexity of solving the ERM problems. By definition, the ERM problem respect to a given dataset D is to minimize the average loss over it. Instead of handling the full dataset all at once, the proposed algorithm starts with a small subset of D and first minimize the ERM problem respect to this subset. After reaching the desired statistical accuracy on this small problem, it doubles the size of the subset and update the problem with the new subset. The strategy repeats such procedure until the full dataset is included. The paper shows an improvement both in theoretical analysis and in experiments. The paper is very clear, the theoretical proof is correct and well presented. I find the paper very interesting and I like the idea of taking into account the statistical problem behind the ERM problem. A question I have is that the algorithm requires to predefine the statistical accuracy and uses it explicitly as a parameter in the construction of the algorithm, is there any way to adapt it by removing its dependency in the algorithm? Because it is demonstrated in the experiments that the choice of such accuracy do influence the performance. (In MNIST, the performance of taking 1/n is better than 1/sqrt{n}) The current algorithm will stop once the predefine accuracy is attained which is eventually improvable by varying it. Besides, I am a bit concerned about the novelty of the paper. As mentioned by the author, there is a big overlap with the reference [12]. The main strategy, including the regularized subproblem and the proposition 1, is the same as in [12]. The only difference is to replace the Newton's method by first order methods and provide the analysis of the inner loop complexity. Overall, I find the idea interesting but the contribution seems to be limited, therefore I vote for a weakly accept.

Reviewer 3



This paper introduce a adaptive method for setting sample size in stochastic methods or converting deterministic methods to stochastic by using mini batch instead of full batch. To compute the mini_batch they consider the estimation error or generalization error in ERM. The main idea of paper comes from ref [6] and [12] of the paper. So the paper tries to generalize these paper's idea. The main question I have is, the V_n is upper bound for estimation error and it depends on the model we use for prediction. So why do you think reaching this level of error is good enough for optimization? what if for the given data set with a rich model we can reach to 0 error? I think this part needs more clarification. All the proves are clear and easy to follow. I believe the experiments do not show the significance of applying your method: 1- For GD or AGD applying your method is like comparing SGD with incremental mini_batch vs GD which is known is SGD with incremental mini-batch is better. 2- based on ref~[1] using growing mini_batch instead of full batch in SVRG helps in convergence. I'd like to see a comparison of your method with other adaptive mini_batch methods to see the actual benefit of applying your method. 1-https://arxiv.org/pdf/1511.01942v1.pdf